# Bounds on the computational complexity of neurons due to dendritic morphology

**Anamika Agrawal**
Center for Data-Driven Discovery
Allen Institute
Seattle, WA 98109
Department of Neurobiology and Biophysics
University of Washington
Seattle, WA 98195
anamika.agrawal@alleninstitute.org

**Michael A. Buice**
Center for Data-Driven Discovery
Allen Institute
Seattle, WA 98109
michaelbu@alleninstitute.org

## Abstract

The simple linear threshold units used in many artificial neural networks have a limited computational capacity. Famously, a single unit cannot handle non-linearly separable problems like XOR. In contrast, real neurons exhibit complex morphologies as well as active dendritic integration, suggesting that their computational capacities outperform those of simple linear units. Considering specific families of Boolean functions, we empirically examine the computational limits of single units that incorporate more complex dendritic structures. For random Boolean functions, we show that there is a phase transition in learnability as a function of the input dimension, with most random functions below a certain critical dimension being learnable and those above not. This critical dimension is best predicted by the overall size of the dendritic arbor. This demonstrates that real neurons have a far higher computational complexity than is usually considered in neural models, whether in machine learning or computational neuroscience. Furthermore, using architectures that are, respectively, more "apical" or "basal" we show that there are non-trivially disjoint sets of learnable functions by each type of neuron. Importantly, these two types of architectures differ in the robustness and generality of the computations they can perform. The basal-like architecture shows a higher probability of function realization, while the apical-like architecture shows an advantage with fast retraining for different functions. Given the cell-type specificity of morphological characteristics, these results suggest both that different components of the dendritic arbor as well as distinct cell types may have distinct computational roles. In single neurons, morphology sculpts computation, shaping not only what neurons do, but how they learn and adapt. Our analysis offers new directions for neuron-level inductive biases in NeuroAI models using scalable models for neuronal cell-type specific computation.

## 1 Introduction

The relationship between neuronal structure and function has received renewed attention due to the growing appreciation of morphoelectric diversity across cell types [Gouwens et al., 2019, 2020, Dembrow et al., 2024, Kalmbach et al., 2021]. Yet, other than classical studies of synaptic integration and circuit motifs, the computational roles of single neurons, shaped by their dendritic morphology, remain underexplored. While traditional models often simplify neurons as linear threshold units, such approximations neglect the rich, nonlinear processing capabilities arising from the spatial and electrical organization of dendritic arbors. Recent experimental evidence demonstrates that

39th Conference on Neural Information Processing Systems (NeurIPS 2025).

dendrites actively integrate inputs in a manner that is both cell-type specific and nonlinear [London and Häusser, 2005]. These findings suggest that dendritic and morphological properties endow neurons with complex computational capabilities, positioning them between simple linear classifiers and deep artificial networks [Cazé et al., 2013, Bicknell and Häusser, 2021].

It has been shown that single neurons can transcend passive linear integration and compute nonlinearly separable functions, such as the XOR function [Gidon et al., 2020] and feature-binding tasks [Bicknell and Häusser, 2021]. However, these studies typically focus on specific nonlinearly separable problems and therefore do not quantify the broader computational limits of single neurons across a range of function complexities. Prior work [Cazé et al., 2013] demonstrated that neurons with local nonlinearities can leverage different integration strategies to solve certain nonlinearly separable tasks. Building on this foundation, we employ Boolean functions to systematically scale the complexity of computational problems with the dimension of input. In addition, we explore specific families of Boolean functions to characterize the capabilities of the dendritic arbor.

In this work, we examine how dendritic morphology constrains the upper bounds of computational complexity achievable by a single neuron. We model neurons as abstract networks of non-linearly integrating dendritic compartments, organized according to archetypal branching architectures. Within this framework, dendritic networks are tasked with learning Boolean functions, allowing precise control over input complexity and objective difficulty. By assessing function realizability across different input dimensions and measures of Boolean function complexity (here, entropy [Mingard et al., 2019] and sensitivity [Kenyon and Kutin, 2004]), we systematically relate morphological features, such as breadth and depth of dendritic trees, to the space of computations that single neurons can perform.

To ground this analysis, we compare two canonical morphologies: (1) a shallow, broad topology reminiscent of basket cells and basal dendritic tufts [Häusser and Mel, 2003, Tzilivaki et al., 2019], and (2) a deep, narrow topology modeled after the apical tufts of excitatory neurons [Häusser and Mel, 2003, Spruston, 2008]. Despite containing the same number of nonlinearly integrating compartments, these architectures exhibit distinct computational strengths: shallow networks excel at learning low-dimensional, low-sensitivity functions, while deep networks generalize more robustly and retrain more efficiently across complex tasks. Importantly, we show that dendritic architectures impose a critical input dimensionality threshold, beyond which the probability of learning a randomly sampled Boolean function collapses sharply. This behavior is reminiscent of phase transitions in circuit complexity (e.g., phase transition in 3-SAT, Karp [2009]) and highlights the intrinsic computational limits imposed by morphology. Furthermore, different dendritic architectures preferentially realize distinct subsets of Boolean functions, suggesting a possible division of computational labor across cell types within neuronal ensembles. Beyond static realizability, we examined how morphology shapes the dynamics of learning itself. We show that apical-like architectures, though less robust initially, retrain and generalize more efficiently once pre-trained, reflecting a flexibility-robustness trade-off that may underlie how different cell types contribute to adaptive computation. Finally, extending this framework, we adapt these morphological motifs to transformer architectures, revealing analogous trade-offs between broad (basal-like) and hierarchical (apical-like) attention patterns, thus bridging biological and artificial systems. By abstracting dendritic trees into structured, deep network-like architectures, our work offers a biologically grounded and tractable framework for understanding how single-neuron computations are shaped, and limited, by morphology. Incorporating such cell-type-specific computations into models of neural circuits opens new avenues for rethinking both theoretical and systems neuroscience paradigms.

## 2 Testing the limits of single neuron computation with abstract models of dendritic morphology

**Abstract Model of a Nonlinear Dendritic Network**

We model a single neuron as a feedforward network comprising dendritic rectification units arranged according to the neuron's dendritic arbor (Fig. 1A). Each compartment $b$ acts as a nonlinear unit in an equivalent multilayer representation:

$$y^b = h^b \sigma \left( \sum_{i=1}^{n} W_i^b x_i - \theta^b \right), \tag{1}$$

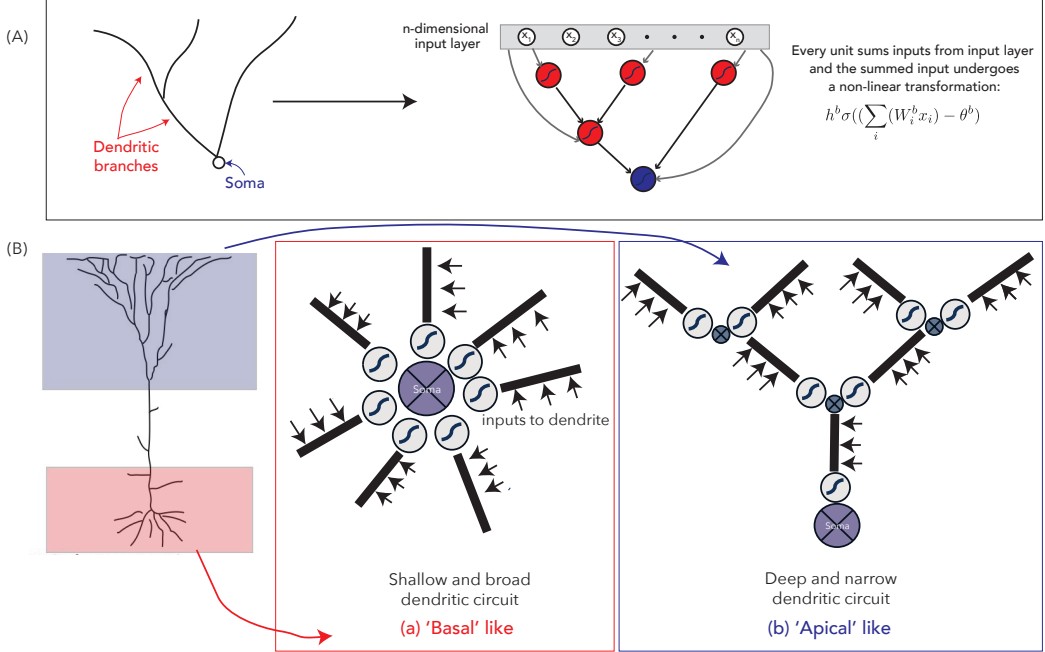

Figure 1: (A) Abstraction of dendritic morphology. (B) Two dendritic architectures: (a) Shallow and broad: all nonlinear compartments connect directly to the soma, resembling basal tufts and enabling parallel processing. (b) Deep and narrow: nonlinear compartments are arranged hierarchically, resembling apical tufts with electrotonically distinct subtrees.

where $y^b$ is the output of branch $b$, $h^b$ and $\theta^b$ are branch-specific gain and threshold parameters, $W_i^b$ are synaptic weights, and $\sigma$ is the logistic sigmoid. If a branch $b'$ also receives inputs from other branches $b$,

$$y^{b'} = h^{b'} \sigma \left( \sum_{i=1}^{n} W_i^{b'} x_i + \sum_b y^b - \theta^{b'} \right). \qquad (2)$$

The soma rectifies its dendritic and direct inputs:

$$y^s = \sigma \left( \sum_{i=1}^{n} W_i^s x_i + \sum_b y^b - \Delta \right), \qquad (3)$$

where $\Delta$ is the somatic threshold. Fig. 1A shows the corresponding network schematic, with input-layer skip connections indicating that all compartments receive direct input.

To test how dendritic arrangement affects computation, we considered two morphological extremes representing trade-offs in circuit breadth and depth under limited total compartments. The *basal/parallel* model has a single hidden layer where all nonlinear compartments connect directly to the soma, resembling basal tufts or basket cells [Tzilivaki et al., 2019, Spruston, 2008]. The *apical* model organizes the same number of compartments into a hierarchical tree, resembling apical tufts of excitatory neurons [Spruston, 2008]. Such an equivalent architecture, if constrained by the number of available dendritic compartments, will be deeper but narrower than the basal model. We typically compared both models with a small dendritic circuit size, $N_{\text{branch}} = 7$ compartments (Fig. 1B).

For comparison, we implemented a linear integration model in which all branches sum inputs linearly before a final somatic nonlinearity:

$$y^{\text{dendrites}} = \sum_{b=1}^{N_{\text{branch}}} \sum_{i} W_i^b x_i, \tag{4a}$$

$$y^s = \sigma\left(\sum_{i=1}^{n} W_i^s x_i + y^{\text{dendrites}} - \Delta\right). \tag{4b}$$

All models, including the linear baseline, used identical input dimensionality and total trainable parameters, differing only in structural organization and presence of intermediate nonlinearities.

**Characterizing the Computational Complexity of a Dendritic Network** The computational complexity of a Boolean function, given a fixed gate set such as AND and OR, is characterized by the number of gates and the minimum circuit depth required to implement it [Wegener, 1987]. Here, we ask: given a specific multilayer network, what limits its ability to realize a family of Boolean functions?

A single neuron receives $N_{\text{dim}}$ input streams that may represent distinct information sources. An input $X_i = 0$ denotes an inactive, noisy source, and $X_i = 1$ an active one; equivalently, 0/1 can be viewed as small versus large EPSPs from distinct synaptic inputs.

Previous work on dendritic learning with increasing input size has focused on memory capacity [Poirazi and Mel, 2001]: the number of patterns learnable within a given $N_{\text{dim}}$. Here, we test instead the ability of a dendritic circuit to implement arbitrary input-output mappings defined by Boolean truth tables, providing a direct measure of computational rather than storage capacity. As shown by Gardner and Derrida [1988], the number of patterns learnable by a linear perceptron scales as $p = aN_{\text{dim}}$ with $a < 1$. Hence, as $N_{\text{dim}}$ increases, the fraction of linearly separable functions $f = p/2^{2^{N_{\text{dim}}}}$ rapidly approaches zero, reflecting the rise in functional complexity with dimensionality. Thus, $N_{\text{dim}}$ serves as an order parameter for scaling task difficulty.

For dimension $N_{\text{dim}}$, there are $x_{N_{\text{dim}}} = 2^{N_{\text{dim}}}$ possible inputs and $n_{\text{func}} = 2^{2^{N_{\text{dim}}}}$ total Boolean functions. Because testing all Boolean functions in high dimensions ($N_{\text{dim}} > 4$ is infeasible, we define three representative families that span the space of task complexity. First, the class of *typical* functions: randomly sampled outputs where each row of the truth table returns 1 or 0 with equal probability ($p = 0.5$). These typical Boolean functions tend to have high sensitivity (defined in the subsequent paragraphs). In the face of extremely large number of possible functions in input dimensions higher than 4, the class of typical functions is to sample the space of "common" functions. We evaluate basal-like, apical-like, and linear-like architectures on sampled typical functions across $N_{\text{dim}}$.

Second, we consider the class of Boolean functions of fixed entropy, defined simply as the fraction min(number of input rows that map to 1, number of input rows that map to 0)/total number of input rows in the truth table [Mingard et al., 2019]. For each input dimensionality, we randomly sampled functions with varying probabilities of outputting 1, thereby covering a range from low-entropy (biased) to high-entropy (balanced) functions.

Finally, we define functions of fixed sensitivity [Rubinstein, 1995, Mossel et al., 2005], where sensitivity quantifies output instability to single-bit flips: The sensitivity of a Boolean function in dimension $n$ is defined as follows:

**Definition 2.1** (Def 1). Let $f : 0,1^n \to 0,1$ be a Boolean function and let $x \in 0,1^n$. The sensitivity of $f$ at $x$ is the number of positions $i \in [n]$ such that flipping the $i^{th}$ bit of $x$ changes the output $f(x)$. The sensitivity $s(f)$ of $f$ is the maximum sensitivity $s(f,x)$ over all points $x$.

Sensitivity increases monotonically with entropy (Supplementary Fig. 5) and scales the difficulty of learning for linear architectures (Fig. 3A).

A low-entropy or low-sensitivity Boolean function can be viewed as one containing non-influential input bits that do not alter the output [Kalai, 2016]. Such functions resemble neural computations in which many inputs are high-dimensional but only a subset carries relevant information, making them effectively lower-dimensional. Because sensitivity increases monotonically with entropy, we used the same set of functions with varying entropy to compute sensitivity, and verified through rejection sampling that both approaches produce comparable distributions.

Together, the random, fixed-entropy, and fixed-sensitivity families span tasks from highly structured to maximally random. For each $N_{\text{dim}}$ = 3-8, we trained all architectures on identical sampled sets (100-500 functions per class) to ensure a fair comparison. Distributions of entropy and sensitivity across function classes are shown in Supplementary Fig. 6.

A Boolean function is considered learned by an architecture if appropriate weights $W_i^b$ and nonlinear parameters $\theta^b, h^b$ can be optimized to map the input $\bar{X} = X_1, X_2, ...X_{N_{dim}}$ to its truth table output $Y^i \forall i \subseteq 1, 2, ....2^{N_{dim}}$, where $i$ denotes the index of the row of the truth table representing the Boolean function. We quantify learning using the Hamming distance between the (thresholded and binarized) dendritic network output and target (see Supplementary Fig. 1). For each function class with $N_{\text{func}}$ functions trained over $n_{\text{trials}}$ random initializations, we compute three learning metrics to test the learnability of the class:

1. The mean Hamming distance between the network output and training target over trials,

$$\hat{HD}^{\text{func}} = \frac{1}{n_{\text{trials}}} \sum_{\text{trials}} HD_{\text{func}}^{\text{trial}} \tag{5}$$

   The mean and variance of this quantity over the sampled functions $N_{func}$ is then visualized.

2. Function learning probability,

$$P_{\text{func}}^{N_{\text{dim}}} = \frac{n_{HD=0}}{n_{\text{trials}}} \tag{6}$$

   is the probability of learning a particular random function and $n_{HD=0}$ is the number of trials where the given function was learned with a Hamming distance $\sim 0$ in $n_{HD0}$. This is the fraction of times the network learned the function perfectly.

3. The 'best-case' learning outcome, i.e. the probability of typical function learning is instead calculated as

$$P_{\text{best-case}}^{N_{\text{dim}}} = \frac{n_{HD=0}^{\text{best}}}{n_{\text{func}}} \tag{7}$$

   where $n_{HD=0}^{best}$ is the number of functions with a Hamming distance of 0 in at least one of the trials tested. This is the fraction of functions for which there is at least one successful trial.

**Training and Optimization Details** We used two complementary optimization schemes: gradient descent and non-gradient based method (DIRECT), to ensure that results were not dependent on the learning algorithm. For all training instances, which entailed learning the function mapping $f : \{0,1\}^n \to \{0,1\}$ in input dimension $n$, the input consisted of the full enumeration of $2^n$ input rows, with the target being the Y-column of the Boolean function mapping.

While training using the gradient-descent based approach, we used Binary Cross-Entropy as the loss function between the predicted and the target outputs during the training. We used an Adam optimizer with a fixed learning rate, with learning rates and epochs tuned according to the input dimension to ensure convergence of loss (ranging from 6e4 epochs with lr = 5e-3 for $N_{dim} = 3$ to 6e6 epochs, lr = 1e-4 for $N_{dim} = 8$. We also ran instances of training with an Adam optimizer with a learning rate scheduler, but that did not impact realizability trends shown in the paper. Parameters ($W_i^b, h^b, \theta^b, \Delta$) were initialized from uniform distributions within bounds chosen to match comparable dynamic ranges across architectures.

To verify that the observed realizability trends were not artifacts of backpropagation, we repeated the analyses using the DIRECT global optimization algorithm [Jones et al., 1993, Gablonsky and Kelley, 2001], which does not rely on gradient information. DIRECT performs a deterministic partition-based search over bounded parameter domains until convergence or a maximum of $5 \times 10^4$ evaluations. Reasonable bounds on the learnable parameters were first set as input parameters for the optimization algorithm. The parameter space was constrained with weights bounded between -20 and 20, branch gains ($h$) between 0 and 10, thresholds ($\theta$) between -1 and 10, and output bias ($\Delta$) between -1 and 10. Both basal and apical cell architectures were trained using mean squared error loss between continuous sigmoid outputs and binary target labels. The DIRECT algorithm performed global optimization without gradients, exploring the 7*d +7 + 7 + 1 - dimensional parameter

space (d input dimensions $\times$ 7 branches + 7 gains + 7 thresholds + 1 bias) to minimize classification error with a maximum of 2,000,000 function evaluations and zero volume tolerance for convergence.

In both cases, training performance was evaluated using Hamming distance between binarized predictions (threshold 0.5) and ground truth binary labels. We trained the 3 dendritic networks (basal, apical, linear) on Boolean functions with input dimensionality varying from $N_{dim} = 3$ to $N_{dim} = 8$. The training time for a given architecture and function over one set of initial conditions (trial) took a maximum of $\sim$ 4 hrs ($N_{dim} = 8$). For multiple trials and functions, we use parallel computing over a cluster to run training on multiple functions and trials simultaneously.

## 3 A phase transition in functional realizability

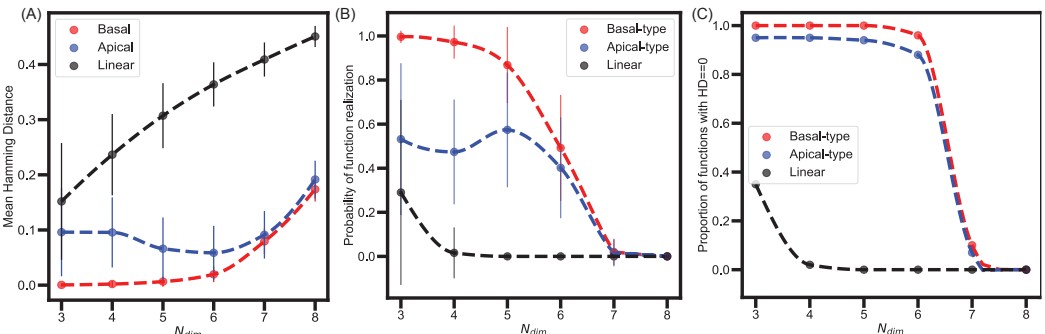

Figure 2: **Phase transition in function realizability as $N_{\mathrm{dim}}$ increases.** (A) Mean Hamming distance across $n_{\mathrm{trials}} = 10$ for $n_{\mathrm{func}} = 100$, calculated as in Eq. 5. (B) Probability of function realization (Eq. 6). (C) Best-case realizability (Eq. 7) for Apical and Basal architectures show transitions at higher $N_{\mathrm{dim}}$ than the linear model.

We first examined how realizability scales with input dimensionality, by sampling $n_{\mathrm{func}} = 1000$ per input dimension. As expected, a purely linear circuit with only a somatic nonlinearity fails to solve most functions even at $N_{\mathrm{dim}} = 3$ (Supplementary Fig. 2), whereas both basal and apical types perform well in this regime. To mitigate stochastic effects, we trained $n_{\mathrm{func}} = 100$ typical functions across $n_{\mathrm{trials}} = 10$ random initializations. The mean Hamming distance across trials (Fig. 2A) increases sharply with $N_{\mathrm{dim}}$, with the linear model failing earliest and both nonlinear architectures maintaining lower error until higher dimensions.

Fig. 2B-C show the corresponding probability of realizability and best-case realizability for typical functions. All architectures exhibit a sharp transition from high to near-zero realizability, analogous to phase transitions in constraint-satisfaction problems. The linear model transitions around $N_{\mathrm{dim}} = 4$, while basal and apical architectures remain functional until $N_{\mathrm{dim}} \approx 7$. This difference highlights how even limited dendritic nonlinearities delay the onset of intractability.

Although a sufficient number of nonlinear compartments ($N_{\mathrm{branch}}$) in dendritic architecture 1(a) can, in theory, approximate any function [Cybenko, 1989], real neurons are constrained by finite compartment number. We find that beyond a critical dimension, the probability of realizing random functions collapses steeply for the basal architecture. This transition was robust to the optimization method: substituting backpropagation with the global DIRECT optimizer [Jones et al., 1993, Gablonsky and Kelley, 2001] produced the same pattern (Supplementary Fig. 4). Beyond this point, functions become effectively unsolvable within a fixed-size nonlinear circuit.

In Supplementary Fig. 3, we analyze how increasing dendritic compartments affects computational capacity. For the basal architecture, size increases by adding branches; for apical, we sample 2-layer tree-like circuits with increasing nodes. In both cases, capacity scales exponentially with compartment number. Because connections are feedforward and sparse, deeper networks trade off breadth, explaining why apical and basal architectures show similar size-cost trade-offs for increasing input dimensionality.

The apical architecture shows lower typical function realizability in low dimensions and greater variability across functions (Fig. 2B), though it matches basal performance in higher dimensions. This

may stem from sensitivity to initial conditions, as uniform weight initialization may disadvantage hierarchical structures. The best-case analysis (Fig. 2C) compensates for this, revealing similar overall trends. We examine this variance further in the next sections.

For some Boolean functions, mismatches in output may be tolerable, e.g., if certain inputs are irrelevant or later corrected by additional signals. Given the high dimensionality of real dendritic inputs, partial learning may suffice. To test this, we relaxed the learning criterion to allow limited output mismatches (Supplementary Fig. 4) and found that in higher dimensions, the apical architecture shows slightly better approximate learning than the basal type.

It is important to distinguish between *trainable realizability* and *best-case realizability*. The former reflects functions successfully learned through gradient-based or DIRECT optimization, whereas the latter quantifies the theoretical capacity of the architecture regardless of optimization path. Across both metrics, the phase transition in realizability remained consistent, confirming that the phenomenon is independent of the training algorithm. Some realizable functions remain difficult to learn in practice due to non-convex loss landscapes or limited parameter resolution; constraints that may be biologically meaningful, as real neurons also compute within restricted parameter regimes. Thus, our empirical results capture both the intrinsic computational capacity of a morphology and the accessibility of that capacity under plausible learning dynamics.

# 4 Morphology-based limits on single-neuron computation

As observed in the error bars in Fig. 2B, sampled functions within a given input dimension $N_{\dim}$ can differ substantially in realizability. To capture this variability and differentiate between functions in the same $N_{\dim}$, we require an additional order parameter - *sensitivity* - which quantifies function difficulty beyond dimensionality alone. To broaden the tested function space, we varied entropy, thereby sampling Boolean functions with diverse sensitivities (definitions in Methods). Supplementary Fig. 6 confirms that entropy-based sampling yields a wide distribution of sensitivities, extending beyond the 100 typical functions used earlier.

Fig. 3B compares learning probabilities for 1000 sampled functions by sensitivity across architectures. Apical and basal models specialize in distinct function classes: functions above the diagonal are better learned by apical cells, while those below favor basal cells. Binning by sensitivity (Fig. 3C) shows that basal cells excel at low-sensitivity functions but degrade as complexity increases, whereas apical cells maintain more uniform, though generally lower, performance across sensitivities. This pattern suggests that apical architectures generalize broadly but imprecisely, while basal architectures learn robustly but with limited transfer. We hypothesize that the apical architecture prioritizes generalization before fine-tuning, whereas the basal architecture learns functions independently and more deterministically. We test this hypothesis in the next section.

# 5 Morphology-based trade-off between computational robustness and flexibility

The apical architecture exhibits lower realizability under random initialization, particularly for low-sensitivity functions. To test whether this reflects distinct learning strategies, we compared retraining flexibility (Fig. 4). Both architectures were first trained on $n_{\text{func}} = 50$ Boolean functions at $N_{\dim} = 5$, with entropy increasing by 0.1 across batches of 10. $N_{dim} = 5$ was chosen to be close to the realizability threshold to ensure learnability. For each function, weights from a successful trial ($HD = 0$) were stored and used for retraining. Each model was then retrained on function $i$ using weights from function $j$, tracking $HD_{j \to i}(t)$ over epochs $t$. The diagonal elements ($j = i$) are zero by design, but pretraining on low-entropy functions ($j = 1$-$10$) accelerates learning for other functions, more strongly in apical than basal architectures, which converges faster than basal across all $j$ (Fig. 4A,B).

We next retrained while fixing branch weights. Branches were indexed by distance from the soma: trunk ($b = 0$), proximal ($b = 1, 2$), and distal ($b = 3$-$6$). Holding the 4 distal branches fixed minimally affected apical retraining but markedly slowed basal retraining, where all branches are equivalent. Additional tests fixing both weights and nonlinearities ($h_b$, $\theta_b$) confirmed that only proximal branches require tuning for apical re-learning (Fig. 7). These findings suggest that apical cells generalize using distal branches trained on simple functions (e.g., AND/OR), which serve as

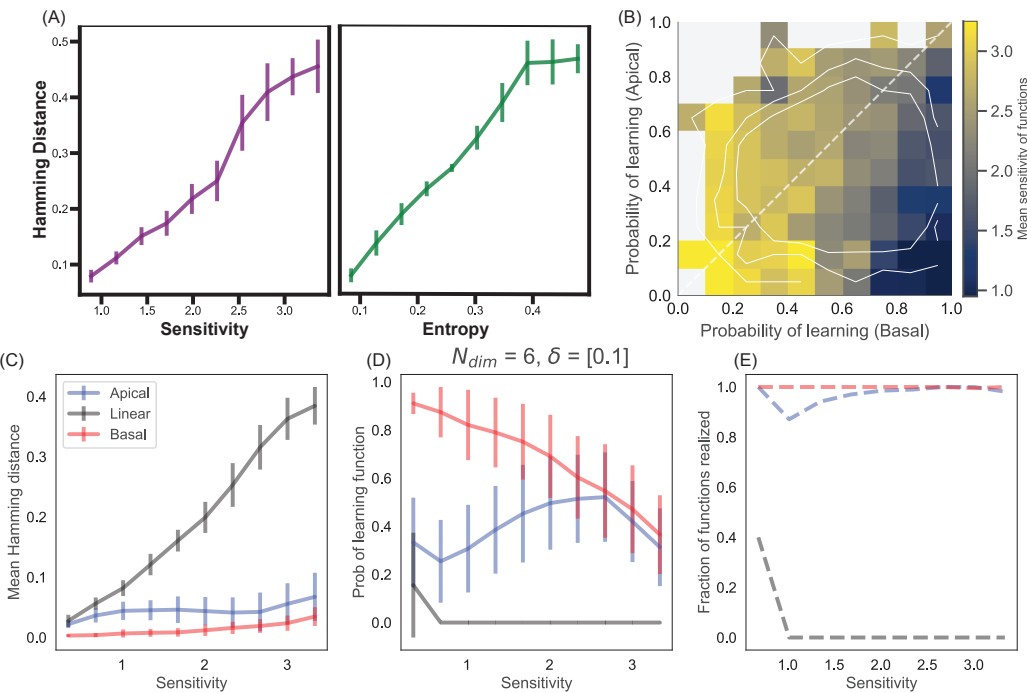

Figure 3: **Learning performance and functional diversity across dendritic architectures.** (A) Mean Hamming distance increases with entropy and sensitivity for linearly integrating neurons, indicating greater difficulty in realizing high-complexity functions. (B) Mean sensitivity of Boolean functions (color scale) in bins of learning probability for basal (x-axis) and apical (y-axis) architectures ($N_{\dim} = 6$). Light-gray regions indicate bins with no sampled functions; white contours (1, 5, 10 functions) mark data density. Functions above the diagonal are preferentially learned by apical architectures, whereas those below are more likely realized by basal architectures. (C-E) Mean Hamming distance, probability of learning, and best-case realizability as a function of Boolean-function sensitivity (mean $\pm$ s.d., $n_{\mathrm{trials}} = 10$). Basal (orange) and apical (teal) architectures diverge in sensitivity dependence, while linear models (gray) fail beyond low-sensitivity regimes. Together, these results show that dendritic morphology shapes the trade-off between robustness and flexibility in single-neuron computation.

reusable building blocks for more complex tasks. In contrast, basal cells lack this generalization and must relearn each function independently (Fig. 4C). Thus, apical architectures are more flexible but less robust, while basal architectures are robust but less adaptable.

## 6 Extending morphological motifs to transformer architectures

To test whether the trade-offs observed in dendritic architectures extend to artificial systems, we adapted the "basal" and "apical" motifs to the layout of attention heads within a transformer model (Supplementary material 1). Both variants contained comparable parameter counts but differed in how attention heads were arranged; broad and shallow versus hierarchical and deep. Across four sequence-learning tasks (copying, reversal, sorting, and pattern completion), broad configurations achieved higher final accuracy, whereas hierarchical configurations converged faster and showed greater improvement during transfer learning. These results suggest that structural organization alone can impose complementary regimes of robustness and flexibility, paralleling the retraining behavior observed in biological dendritic networks.

## 7 Discussion

Our results demonstrate that dendritic morphology is not merely a structural feature but a fundamental determinant of a neuron's computational capacity. By abstracting dendritic trees into simple

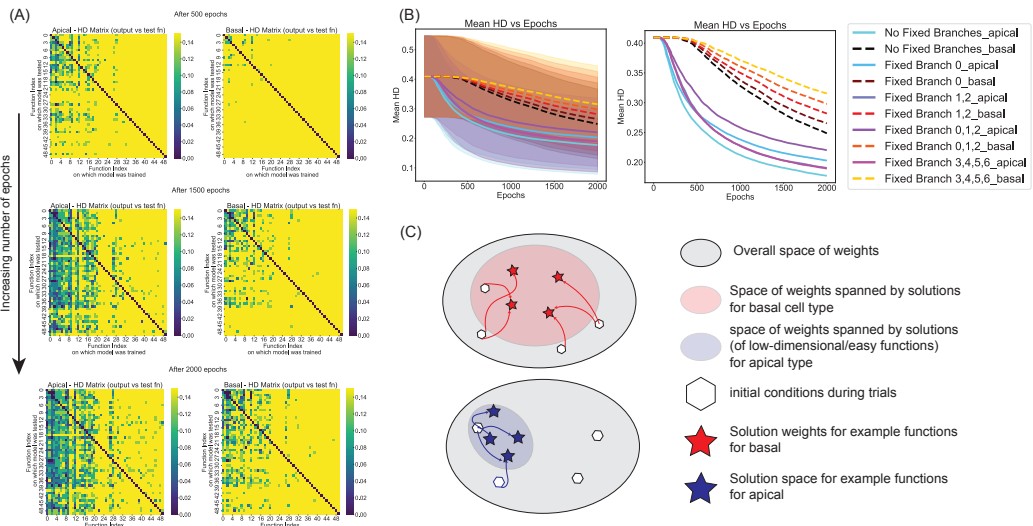

Figure 4: **Apical architectures enable flexible retraining.** (A) Hamming distance $HD_{j \to i}(t)$ between pre-trained function $j$ and test function $i$ shows faster convergence for apical cells, especially when pre-trained on low-entropy functions ($j = 1$-10). (B) Mean Hamming distance under different branch-freezing conditions; apical performance remains stable when distal branches (3-6) are fixed, indicating their role in generalization. (C) Schematic: basal architectures are robust to initialization, whereas apical architectures are flexible once pre-trained. Branch-specific freezing parallels dendritic disinhibition that gates plasticity and context-dependent learning [Canto-Bustos et al., 2022, Onasch et al., 2025].

| Architecture | Critical Dim. | Sensitivity Preference | Robustness | Retrainability | Generalization |
|---|---|---|---|---|---|
| Basal-like | High | Low-sensitivity favored | High (init-cond tolerant) | Low | Learns functions independently |
| Apical-like | High | No specific preference | Low | High | Shares features via distal branches |
| Linear | Very low | Only very low-sensitivity | Poor | Poor | Purely linear; minimal computation |

Table 1: Summary of computational properties across dendritic architectures.

models, we show that neural geometry imposes strict bounds on the input-output functions a single neuron can realize. While nonlinear dendritic integration greatly expands computational capabilities beyond linear models, this expansion is not unbounded. Beyond a critical dimension, the probability of learning random Boolean functions collapses. This phase transition mirrors phenomena in circuit complexity theory, such as the 3-SAT satisfiability transition.

Importantly, dendritic architectures have computational trade-offs. 'Basal-like' shallow, broad architectures favor robust, condition-insensitive computation, reliably learning simple functions across diverse initializations. In contrast, 'apical-like' deep, narrow architectures favor flexibility: though initially less robust, once properly initialized, they retrain and generalize across a broader range of functions with fewer adaptations. These complementary properties suggest a natural division of computational labor across cell types: robust processors for stable operations and flexible processors for dynamic reconfiguration.

**Neuroscience implications** Our results complement and extend a growing body of experimental and theoretical work showing that dendritic morphology shapes how neurons compute. Human layer 2/3 pyramidal neurons exhibit graded calcium-mediated dendritic spikes that enable single neurons

to solve linearly non-separable tasks [Gidon et al., 2020], while morphologically distinct apical and basal arbors in rodent pyramidal cells support compartmentalized integration during behavior [Otor et al., 2022]. In hippocampal CA1 and CA3 neurons, distal dendrites contribute to place field remapping and flexible memory encoding [Li et al., 2023, OHare et al., 2025], suggesting that dendritic subregions encode functionally distinct computations shaped by morphology and context. More generally, studies such as Beniaguev et al. [2021], de Las Casas et al. [2025] and recent reviews [Tye et al., 2024, Makarov et al., 2023, Payeur et al., 2019] emphasize that nonlinear dendritic subunits enhance representational dimensionality and learning flexibility. Our framework provides a complementary, function-space perspective on these findings, formalizing how dendritic structure constrains the computational regimes accessible to single neurons.

These principles have direct implications for cortical circuit organization. Excitatory neurons with elaborate apical tufts, such as layer 5 pyramidal neurons, may leverage their architecture to flexibly adapt to changing cognitive demands, whereas inhibitory or smaller pyramidal neurons with simpler dendritic arbors promote stability and reliability. Recent work suggests that these structural specializations are coupled to distinct modes of plasticity: in mouse motor cortex, apical and basal compartments follow different learning rules [Wright et al., 2025]. The trade-off between flexibility and robustness we observe echoes these findings. Likewise, our branch-specific weight-freezing experiments parallel dendritic disinhibition mechanisms that gate plasticity and promote context-dependent learning [Canto-Bustos et al., 2022, Onasch et al., 2025]. Modeling efforts that integrate such morphological and compartment-specific constraints could therefore yield more realistic predictions of how cortical circuits balance adaptability with stability.

**Network-level implications** Beyond the single-cell level, these results have direct implications for network complexity. Neurons with greater intrinsic computational capacity reduce the number of units required to represent a given function, analogous to how richer gate sets affect scaling constants in circuit complexity classes such as NC or AC. Morphological constraints at the single-neuron level could therefore influence the size, efficiency, and modularity of biological circuits. In this view, neuronal diversity does not merely reflect biological variation but an optimization of computational trade-offs; balancing capacity, robustness, and energy cost across scales of organization.

**Relevance to NeuroAI** Our results also suggest opportunities for artificial architectures. We demonstrate that the same structural trade-offs observed in neurons also arise in artificial architectures. Such dendritic-inspired diversity could help artificial systems balance stability and continual learning. Future work incorporating these hybrid architectures may provide a path toward more adaptive, resource-efficient models bridging biological and artificial intelligence.

**Limitations and outlook** Our framework simplifies many aspects of neuronal computation. We focus on feedforward architectures with rectified nonlinearities, abstract away spiking dynamics, recurrence, and inhibition, and use Boolean function learning as a proxy for computational complexity. Our analysis is phenomenological; the observed phase transition in realizability has not yet been formally derived from first principles, and its theoretical characterization within circuit complexity or learning theory remains open. Experimental validation, for instance through dendritic imaging or targeted perturbations that test compartment-specific flexibility and robustness, will be essential for linking these abstract results to biological neurons. Nevertheless, the qualitative insights remain robust; morphology directly shapes the computational strategies available to a neuron.

By linking dendritic morphology to single-neuron computational limits via deep network architectures, we provide a tractable, scalable approach bridging cellular neuroscience with theoretical frameworks from machine learning and computational complexity. Future work extending this to ensembles of diverse neurons could reveal how local computational trade-offs influence circuit-level dynamics, shaping cognition, learning, and dysfunction. In short, morphology sculpts computation, shaping not only what neurons do, but how they learn and adapt. Table 1 summarizes how dendritic architectures impose distinct trade-offs between robustness, flexibility, and sensitivity, suggesting that evolution has optimized not just wiring efficiency but computational strategy at the cellular level.

## Acknowledgements

This work was conducted at the Allen Institute and the University of Washington, whose support is gratefully acknowledged. Fellowship support for A.A. was provided by the Shanahan Family Foundation. Computational resources were provided by the Allen Institute High Performance Computing (HPC) team. We thank Kaspar Podgorski, Frederick Rieke, Forrest Collman, Casey Schneider-Mizell, Lukasz Kusmierz, and Anton Arkhipov for valuable discussions and feedback. Code associated with this work will be available at `github.com/AnamikaAg/dendCompPaperCode`

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
