# OpenReview forum: "Bounds on the computational complexity of neurons due to dendritic morphology"
_NeurIPS.cc/2025/Conference — NeurIPS 2025 poster_

### Official Review · Reviewer_qnNj · 2025-06-17

**Clarity:** 2
**Significance:** 2
**Originality:** 2
**Rating:** 4
**Confidence:** 3

**Summary:**

This paper investigates the computational and training capabilities of alternatives to standard linear threshold units, to potentially explain the difference in learning ability of standard neural network architectures and human brains. Specifically, it explores two types of replacement units that differ in topology: apical-like units, which are thin and deep, and basal-like units, which are broad and shallow.

To evaluate the computational complexity of these configurations, the authors analyze the types and proportions of random Boolean functions that each architecture can learn. They focus on functions grouped by similar sensitivity and entropy levels. Their findings suggest that architectural topology significantly influences learning behavior, and that there are non-trivially disjoint sets of learnable functions by each type of neuron.

Basal-like architectures exhibit high computational capacity and show a strong preference for learning functions with high sensitivity. However, they are not very robust and can only be retrained effectively on functions with very low sensitivity. Despite this, they generalize well and tend to learn functions independently, without much feature sharing.

Apical-like architectures, in contrast, have lower computational capacity and do not show a strong preference for sensitivity levels. They are more robust, particularly tolerant to variations in initial conditions, but they perform poorly in terms of retrainability and generalization. These architectures tend to share features across functions via their distal branches.

Linear architectures are the simplest of the three, with minimal computational power. They have low sensitivity, poor robustness, and limited retrainability. However, they generalize relatively well. Their computation is purely linear, offering little in terms of complex function learning.

**Questions:**

It would strengthen the paper to include a discussion on how the insights from this analysis could inform the design or modification of existing training architectures—particularly in widely used models like transformers for language tasks. For instance, if certain topologies (e.g., basal-like) are shown to be more robust or generalizable for specific classes of Boolean functions, could similar structural biases be introduced into transformer blocks to improve their inductive biases or training dynamics?

More concretely, it would be helpful to know:

1. Whether these findings suggest new ways to structure attention heads or feedforward layers.
2. If certain types of functions (e.g., high-sensitivity ones) are better captured by specific architectural motifs.
3. How these insights might guide the design of hybrid models or inform pruning strategies.

A forward-looking discussion on these implications would make the work more actionable and relevant to the broader machine learning community.

**Ethical Concerns:**

["NO or VERY MINOR ethics concerns only"]

**Final Justification:**

My questions have been addressed, and I find the results interesting—so I support accepting the paper. That said, it would be even stronger if the authors showed how their findings could guide the design of modern architectures. They’ve agreed to add such a discussion in the final version, and because I haven’t seen that yet, I’m updating my score from a borderline reject to a borderline accept.

**Limitations:**

Yes, but it might be better to have a separate "Limitations" section in the paper.

**Quality:**

2

**Strengths And Weaknesses:**

Strengths

This paper presents an interesting and thoughtful analysis. While the conclusions may not be particularly surprising—given the extensive prior work on architectural variations—it’s valuable to see such a detailed and focused study in the context of Boolean functions.

Weaknesses

1. The notion of entropy used in the paper could benefit from a clearer definition. In the context of Boolean function analysis (e.g., [1]), terms like Fourier entropy have specific meanings that may not align with the usage here.

2. The training procedure should be described in more detail in the main body. Different training setups can lead to different local minima, which may affect the learnability of certain functions.

I also feel the analysis, while interesting, is somewhat simplistic. It would help to know to what extent and in what contexts the conclusions of this could be further generalized and will transfer. A discussion on this will help strengthen the paper.

Reference
[1] Ryan O'Donnell, Analysis of Boolean Functions. arXiv:2105.10386

---

> ### Author Rebuttal · Authors · 2025-07-31
>
> Thank you for your thoughtful and detailed review. We appreciate your comments and suggestions, especially regarding clarity around terminology and the relevance to machine learning architectures. Below, we respond to your main concerns and describe the changes we’ve made in the revised manuscript.
>
> ## Entropy definition
> You are absolutely right that the term “entropy” carries specific meanings in the context of Boolean function analysis, such as Fourier entropy. In our paper, we use entropy as defined by Mingard (2019), which is calculated over the output column.  This definition maps to Shannon Entropy monotonically but non-linearly (ranges from 0 to 0.5 instead of 0 to 1). For a Boolean function with binary outputs, this corresponds to the uncertainty over whether the function returns 0 or 1  across the set of input configurations. We have now clarified this definition explicitly in the main text and noted that it differs from Fourier entropy as discussed in the reference by O'Donnell.
>
> We also added a brief comment in the Discussion to highlight that future extensions could explore spectral characteristics of function realizability. This may be particularly useful when comparing with architectures that are biased toward or against specific frequency components in their learned representations.
>
> ## Training procedure
> Thank you for flagging this. We apologize for the lack of detail.  We will follow your suggestion and add clear statements about training methods, which we detail below.  It is important to note that we used two distinct approaches, gradient descent, as well as a non-gradient based method (DIRECT) in order to test for limitations of the learning algorithms.
>
> For all training instances, which entailed learning the function mapping $f : \\{0,1\\}^n \rightarrow \\{0,1\\}$ in input dimension $n$, the input consisted of the full enumeration of $2^n$ input rows, with the target being the Y-column of the boolean function mapping.
> While training using the gradient-descent based approach, we used Binary Cross-Entropy as the loss function between the predicted and the target outputs during the training. We used an Adam optimizer with a fixed learning rate, with learning rates and epochs tuned according to the input dimension to ensure convergence (ranging from 6e4 epochs with lr = 5e-3 for $n_{dim} =  3$ to 6e6 epochs, lr =  1e-4 for $n_{dim} =8$. We also ran instances of training with an Adam optimizer with a learning rate scheduler, but that did not impact realizability trends shown in the paper.
>
> For the DIviding RECTangles (DIRECT) optimization, we used Scipy's implementation which was based on Gablonsky et al 2001. Reasonable bounds on the learnable parameters were first set as input parameters for the optimization algorithm. The parameter space was constrained with weights bounded between -20 and 20, branch gains ($h$) between 0 and 10, thresholds ($\theta$) between -1 and 10, and output bias ($\Delta$) between -1 and 10. Both basal and apical cell architectures were trained using mean squared error loss between continuous sigmoid outputs and binary target labels. The DIRECT algorithm performed global optimization without gradients, exploring the 7*d +7 + 7 + 1 - dimensional parameter space (d input dimensions × 7 branches + 7 gains + 7 thresholds + 1 bias) to minimize classification error with a maximum of 2,000,000 function evaluations and zero volume tolerance for convergence. Training performance was evaluated using Hamming distance between binarized predictions (threshold 0.5) and ground truth binary labels.
>
> For a broad overview on the evaluation of realizability of a boolean function by a given architecture, we point the reviewer to Supplemental Figure 1.
>
> By reporting both training success rate and best-case realizability, we are able to distinguish between functions that are simply hard to train and those that are likely not realizable under the given morphology. We will revise Section 2 to make this workflow clearer.
>
> ## On the simplicity of the analysis and relevance to real-world models
>
> We agree that the framework is abstract by design. Our goal was to create a clean setting in which we could isolate the influence of dendritic topology on functional realizability and adaptability. That said, your question about how these insights transfer to modern architectures like transformers is well taken. It also aligns with questions raised by other reviewers, and we've started investigating this directly.
>
> As a preliminary step, we adapted our “broad” and “hierarchical” architectural motifs to the layout of attention heads within a transformer. The total number of heads was held constant, but arranged in different patterns. Hierarchical configurations were deep and narrow, while broad ones were shallow and wide. We evaluated these models on four sequence-learning tasks: copying (short-term memory), reversal (long-range dependency), sorting (comparison), and pattern completion (reasoning). The training details are as follows:
> We compiled results from training with 8 tree attention architectures (4 hierarchical, 4 broad) across 4 sequence prediction tasks (copying, reversal, sorting, pattern completion). Models were trained with model dimension $d_{model}=128$, 3 layers, sequence length=12, vocabulary size=20, for 200 epochs using an Adam optimizer with learning rate=0.0005. Convergence was measured as the average number of epochs to stable validation loss. Transfer learning was evaluated via fine-tuning pre-trained models on new tasks compared to training from scratch, with negative values indicating performance degradation relative to baseline.
>
> The exploratory analysis is summarized in the tables below.
>
> ### Table: Tree Attention Architecture Performance Comparison
> ### Results from 4 sequence prediction tasks. Convergence and transfer metrics averaged by architecture type.
> | **Architecture** | **Type**     | **Avg Loss** | **Task Wins** | **Rank** |
> | ---------------- | ------------ | ------------ | ------------- | -------- |
> | \[15, 1]         | Broad        | **0.2004**   | 2/4           | 1        |
> | \[19, 1]         | Broad        | 0.2296       | 0/4           | 2        |
> | \[9, 1]          | Broad        | 0.2394       | 1/4           | 3        |
> | \[13, 1]         | Broad        | 0.2410       | 1/4           | 4        |
> | \[12, 6, 2]      | Hierarchical | 0.7617       | 0/4           | 5        |
> | \[8, 4, 2]       | Hierarchical | 0.7625       | 1/4           | 6        |
> | \[10, 5, 1]      | Hierarchical | 0.7687       | 0/4           | 7        |
> | \[6, 3, 1]       | Hierarchical | 0.7796       | 0/4           | 8        |
>
> ### Architecture Type Summary
>
> | **Metric**       | **Hierarchical** | **Broad**    | **Winner**   |
> | ---------------- | ---------------- | ------------ | ------------ |
> | Average Loss     | 0.7681           | **0.2276**   | Broad        |
> | Task Win Rate    | 25%              | **75%**      | Broad        |
> | Convergence Time | **148.9 epochs** | 165.7 epochs | Hierarchical |
> | Transfer Benefit | **-11.5%**       | -21.5%       | Hierarchical |
>
> Some early trends that emerge:
> - Broad architectures performed better overall across most tasks. This is consistent with the idea that parallel architectures are better suited for handling high-sensitivity or high-throughput inputs.
> - Hierarchical models, while performing worse on final accuracy, converged faster during training. They also performed significantly better during transfer learning, particularly when fine-tuned across tasks.
>
> These findings parallel what we observe in our main paper. Shallow models perform well on fixed tasks but generalize less effectively. Deeper models, although less expressive in some regimes, are more flexible in how they adapt and reuse internal structure. We will include a summary of these early experiments in the Discussion. Although still exploratory, we believe this direction is promising and provides a bridge between our abstract neuron model and architectural questions in deep learning.
>
> ## On hybrid models and broader implications
>
> Your suggestion to consider implications for architectural motifs and pruning strategies is very much appreciated. We have added a few lines to the Discussion exploring how morphological constraints might inspire hybrid or modular designs. For example, one could imagine incorporating a mix of broad and hierarchical processing units in a transformer to balance robustness with adaptability. Similarly, knowing that certain architectures favor specific function classes might help guide structured pruning approaches or inform initialization strategies.
>
> ## Limitations
>
> We agree that the limitations of the current work should be stated more directly. We've added a brief "Limitations" section in the main paper that highlights some of the key boundaries of the current study, such as the use of synthetic functions, the limited input dimensionality, and the need for further validation in real-world tasks.
> Thank you again for your constructive feedback. We've clarified terminology, improved the description of our methods, and added a broader discussion of how these findings may transfer to machine learning models.

---

> ### Comment · Reviewer_qnNj · 2025-08-03
>
> Thank you for your detailed response. I will adjust the score in light of this.

---

### Official Review · Reviewer_nE68 · 2025-06-18

**Clarity:** 2
**Significance:** 3
**Originality:** 3
**Rating:** 4
**Confidence:** 4

**Summary:**

The paper investigates how the dendritic morphology of single neurons constrains their computational capacity, using abstract neural network models inspired by biological dendritic architectures. The work discusses the computational capability of two types of dendritic morphology by using three types of Boolean functions.
The authors discover a phase transition in functional realizability with increasing input dimensionality. They show that this transition is strongly influenced by the dendritic architecture and the number of nonlinear compartments.
These findings suggest that morphological features of neurons impose structural inductive biases that shape their computational strategies. The work opens up new directions for incorporating biologically grounded neuron models into machine learning, particularly for tasks involving continual adaptation and resource efficiency.
They reveal also that “basal” architectures are more robust and perform well on low-complexity functions. “apical” architectures are more flexible, better at generalization and retraining, especially after pretraining on simple tasks. These findings suggest that morphological features of neurons impose structural inductive biases that shape their computational strategies. The work opens up new directions for incorporating biologically grounded neuron models into machine learning, particularly for tasks involving continual adaptation and resource efficiency.

**Questions:**

1. The authors present abstract models for the linear, apical, and basal architectures. However, the specific structural differences among these models are insufficiently described, making it difficult to align the experimental results with the stated conclusions. For example, the “linear” variant appears to be a degenerate form of the basal model without activation functions, yet this distinction is not clearly articulated. Furthermore, to ensure a fair comparison, the number of parameters in the linear model should be matched with those in the apical and basal models.
2. While the authors explore the impact of dendritic morphology on Boolean functions across input dimensions ranging from 3 to 8, this dimensional range is quite limited compared to real-world applications. It remains unclear how the findings scale to higher-dimensional inputs, which are more representative of biological and artificial neural systems.
3. The authors introduce three types of Boolean functions, i.e., typical, fixed entropy, and fixed sensitivity, but do not clearly explain how these function sets are used to evaluate model performance, nor how the ratios or distributions of these sets are determined. This obscures the interpretability and reproducibility of the experimental setup.
4. The paper reports an interesting phase transition in function realizability at a certain critical input dimension, which appears to correspond roughly to the number of dendritic branches. However, it is unclear whether this critical dimension can be formally predicted or calculated based on model parameters. Providing a theoretical estimate or analytical expression for this threshold would significantly strengthen the paper’s contributions.
5. In Section 5, the authors investigate retraining dynamics by freezing the weights of specific dendritic branches. However, the branch indexing scheme is not clearly defined, making it difficult to interpret which components are being constrained. This lack of clarity weakens the conclusions regarding retraining flexibility across architectures.

**Ethical Concerns:**

["NO or VERY MINOR ethics concerns only"]

**Final Justification:**

Although the author's response addressed some detailed issues, the overall quality of the work has not improved significantly. Therefore, I maintain my original rating.

**Quality:**

2

**Strengths And Weaknesses:**

The authors innovatively employ three classes of Boolean functions to evaluate the computational capabilities of different dendritic neuron models (apical and basal), and offer a compelling discussion on the robustness and flexibility trade-offs between architectures. However, several concerns remain:
1. The modeling details are insufficiently specified, making it difficult to assess the fairness and reproducibility of the architectural comparisons.
2. The description of the Boolean functions is vague, and it is unclear how representative or diverse these functions are.
3. While the discovery of a critical dimension associated with a phase transition is intriguing, the experiments are conducted on a relatively small function set, and the lack of accompanying theoretical analysis limits the generality of the findings.
4. Although the authors provide empirical support for their claims, the scalability and practical relevance of the proposed approach remain uncertain due to the restricted experimental scope.

---

> ### Author Rebuttal · Authors · 2025-07-31
>
> Thank you for your thoughtful and constructive feedback. We appreciate your recognition of the core ideas, especially the link between dendritic morphology and architectural trade-offs, and we agree that there are places where the exposition could be clearer. Below, we address each of your major concerns in turn, and we plan to revise the manuscript accordingly.
>
> ## Modeling details
> Thank you for pointing out that our modeling details are insufficiently fleshed out.  We apologize for the lack of detail.  We will follow your suggestion and add clear statements about training methods, which we detail below.  It is important to note that we used two distinct approaches, gradient descent, as well as a non-gradient based method (DIRECT) in order to test for limitations of the learning algorithms.
>
> For all training instances, which entailed learning the function mapping $f : \\{0,1\\}^n \rightarrow \\{0,1\\}$ in input dimension $n$, the input consisted of the full enumeration of $2^n$ input rows, with the target being the Y-column of the boolean function mapping.
> While training using the gradient-descent based approach, we used Binary Cross-Entropy as the loss function between the predicted and the target outputs during the training. We used an Adam optimizer with a fixed learning rate, with learning rates and epochs tuned according to the input dimension to ensure convergence (ranging from 6e4 epochs with lr = 5e-3 for $n_{dim} =  3$ to 6e6 epochs, lr =  1e-4 for $n_{dim} =8$. We also ran instances of training with an Adam optimizer with a learning rate scheduler, but that did not impact realizability trends shown in the paper.
>
> For the DIviding RECTangles (DIRECT) optimization, we used Scipy's implementation which was based on Gablonsky et al 2001. Reasonable bounds on the learnable parameters were first set as input parameters for the optimization algorithm. The parameter space was constrained with weights bounded between -20 and 20, branch gains ($h$) between 0 and 10, thresholds ($\theta$) between -1 and 10, and output bias ($\Delta$) between -1 and 10. Both basal and apical cell architectures were trained using mean squared error loss between continuous sigmoid outputs and binary target labels. The DIRECT algorithm performed global optimization without gradients, exploring the 7*d +7 + 7 + 1 - dimensional parameter space (d input dimensions × 7 branches + 7 gains + 7 thresholds + 1 bias) to minimize classification error with a maximum of 2,000,000 function evaluations and zero volume tolerance for convergence. Training performance was evaluated using Hamming distance between binarized predictions (threshold 0.5) and ground truth binary labels.
>
> For a broad overview on the evaluation of realizability of a boolean function by a given architecture, we point the reviewer to Supplemental Figure 1.
>
> ## Model structure and fair comparisons
>
> We agree that the distinctions between the linear, apical, and basal models could be better described. In the revised manuscript, we plan to clarify that the “linear” model is not simply a degenerate case of the basal architecture. All models - including the linear baseline - use the same total number of trainable parameters, and each receives the same number of input features. What differs is how those parameters are arranged structurally, and whether intermediate nonlinearities (e.g., dendritic compartments) are present. We plan to include a table/schematic to make this comparison explicit.
>
> ## Boolean function diversity and usage
> Thank you for pointing out that the description of the different Boolean function classes was not sufficiently clear. We’ve now clarified the definition of each class as well as their rationale in the revised Methods section. Briefly:
> - Random Boolean functions:  We devise a truth table that picks the output from $\{0,1\}$ with equal probability for each row of the truth table.  Since this procedure equally weights each function, these functions will typically saturate known theoretical bounds on circuit complexity.  Namely, they typically require an exponential number of gates.  From this perspective, they can be considered the ``hard" functions.  These functions tend to have high entropy and high sensitivity, making them difficult to implement for any fixed architecture. We have included a plot in the supplement that visualizes the sensitivity of random boolean functions, where we show that the sensitivity values are close to the theoretical maximum sensitivity of boolean functions.
>
> - Fixed Entropy functions:  Entropy of a boolean function is defined in the paper as the fraction min(number of input rows that map to 1, number of input rows that map to 0)/total number of input rows in the truth table [Mingard et al., 2019]. These functions are generated randomly by mapping the complete set of input columns to the output column with the output column generated using random sampling from the possibilities [0,1] with probability (p, 1-p), where p is the desired entropy. This family of functions provides for a parametric way to vary function complexity.  The lowest entropy function is the constant function, whereas the highest is the Random Boolean functions.
>
> - Fixed Sensitivity functions: The sensitivity of a boolean function is calculated according to definition 2.1 in the paper. We generate this class of functions using rejection sampling to select for functions with sensitivity in a given range near the required sensitivity. This family of functions provides for a parametric way to vary the degree to which the function output is sensitive to change in the input.
>
> - Sensitivity and entropy are related, which allows us to sample high-sensitivity functions by sampling in the subspace of high entropy functions. We show this in Supplemental Figure 6
>
> - In each experiment, we sample a large number of functions from each class at a given input dimensionality (e.g., 100-500 per class), and all architectures are trained and tested on the same sampled sets to ensure fairness.
>
> - We have now included summary statistics and visualizations of sensitivity and entropy distributions for each function class.
>
> ## Theoretical analysis
>
> We agree that a complete theoretical account of this phase transition would strengthen the paper and we have been working towards that goal.  We can offer a proof that there exists a dimension above which typical functions will be impossible to learn.  Briefly, Shawe-Taylor, et al (1992) showed that the class $NN_k$ is contained in $AC_k$.  $NN_k$ is the class of polynomial size neural networks of finite depth (with fixed bit precision in the weights).  $AC_k$ is the familiar ``Alternating Circuit" class of Boolean functions of finite depth and polynomial number of gates.  Given known bounds on the complexity of generic Boolean functions (namely, they require an exponential number of gates), this implies that there exists some dimension beyond which most functions will not be implementable by a given architecture of neural network.  Our empirical results quantify this for specific architectures.  This phase transition appears similar to that observed for 3SAT as a function of the ratio of literals to clauses.  Intuitively, adding constraints that outnumber available configuration variables leads to unsolvability.  We will add this argument to the revised manuscript.
>
> ## Scalability to high-dimensional inputs
> This is an important question. While our current results focus on input dimensions up to 8, the main limitation isn't in the models but in the function space, specifically, the exponential growth in the number of Boolean functions makes sampling and evaluation difficult beyond that point. That said, we're actively extending the framework to larger inputs using structured function classes with lower entropy, such as compositional rules or localized feature detectors, which are more representative of real neural coding regimes. We've added a paragraph outlining this extension and our ongoing work in this direction.
>
> ## Clarifying retraining results and labeling
>
> Thanks for pointing out the ambiguity in Section 5. We will clarify the branch indexing scheme in the text and caption, and add labels in Figure 4 to make it clearer which dendritic subtrees were frozen during retraining. The revised figure and caption will now make this explicit.
>
> ## Summary
> We've revised the manuscript to improve clarity on model design, training setup, function class definitions, and figure labeling. We also now include a clearer discussion of theoretical and experimental implications, and we've flagged ongoing work that will help address scalability and generalization beyond the current function set.
>
> Thank you again for the detailed feedback. It’s helped us make the manuscript more precise, and we hope the revised version better communicates the motivation and scope of our work.

---

### Official Review · Reviewer_BeVr · 2025-06-30

**Clarity:** 3
**Significance:** 2
**Originality:** 2
**Rating:** 3
**Confidence:** 3

**Summary:**

This work explores how dendritic morphology constrains the computational capacity of individual neurons by modeling them as feedforward networks with non-linear dendritic compartments. The authors compare basal-like and apical like morphology vs point neuron architecture to  analyze which classes of Boolean functions are realizable by each. The study reveals a sharp phase transition in function learnability with increasing input dimensionality, provides insights into architecture-specific trade-offs between robustness and flexibility, and connects findings to implications in neuroscience and artificial intelligence.

**Questions:**

While this simplified analysis can help us gain insight on the underlying mechanism that might offer advantage for dendritic architecture.  I suggest authors to show that their theory is still relevant under realisitc datasets and model of more than a single neuron.

**Ethical Concerns:**

["NO or VERY MINOR ethics concerns only"]

**Final Justification:**

After review the rebuttal and discussion among authors and reviewers, I decide to maintain my current evaluation. The main reason is I believe the very low dimensionality used in this study makes it not convincing.

**Limitations:**

No. Please see Strengths And Weaknesses.

**Quality:**

2

**Strengths And Weaknesses:**

Strength:
The work systemtically inspects function realizability across dimensions and complexities. This can help us gain better understanding the role played by dendritic architecture.
The distinction between apical and basal architectures is a important contribution.

Weakness:
The assumption of data input dimensionality is very far from reality (N_dim ≤ 8). The study assume a single neuron need to be responsible for seperating input patterns invidivually. In reality, neurons only receives very spare (and very high dimesion) inputs. The information is likely encoded in superposition manner. Likely the learning rely on the coordination of many neurons.

---

> ### Author Rebuttal · Authors · 2025-07-31
>
> Thank you for your review and for raising important questions about dimensionality and realism. These are fair concerns, and we appreciate the chance to clarify our framing and explain where we see both the limits and utility of our approach.
>
> ## On the question of dimensionality
> You're absolutely right that real neurons receive high-dimensional, sparse inputs, and that their computation typically occurs in coordination with many other neurons.  It remains an open question whether neurons can non-trivially coordinate these inputs, e.g. compute non-linearly separable functions of multiple inputs.  This can include a relatively simple function such as XOR, a function of only two inputs, or more complicated functions such as feature binding.  The dimensionality of inputs considered in these cases is not equal to the total dimensionality of the inputs a real neuron receives.
>
> One reason to consider relatively low-dimensional inputs is tractability.  Another is the observation that inputs to neurons are not typically highly synchronous or highly correlated.  Analyzing random Boolean functions lets us consider functions of a kind of "worst-case" complexity and demonstrate that there is an upper bound here.  Random Boolean functions are typically "hard". While the number of possible functions grows rapidly with input size, the complexity of randomly sampled functions is already very high at low dimensions. These functions tend to have high entropy (as defined in the paper) and high input sensitivity, which makes them difficult to learn regardless of architecture. In this sense, they serve as a kind of stress test. If a neuron can't learn a function in this regime, it likely wouldn't scale well to even more complex ones.
>
> Similarly, realistic computations are often low-dimensional in practice. Even though a neuron may receive hundreds of inputs, many known examples of single-neuron computation (such as coincidence detection, sequence recognition, or spatially restricted integration) can be mapped to low-dimensional, low-sensitivity function classes. We are currently preparing a small table to clarify this point. It will show examples of computations that neurons are thought to perform and how they map onto specific Boolean function types.
>
> ## On the value of studying single-neuron realizability
>
> Thank you for the suggestion to discuss the network-level implications of this work.  While we focus on single neuron computational complexity, these results have immediate impact on the complexity of entire circuits.  A similar question arises in the domain of circuit complexity when considering Boolean circuits.  In that case, one specifies a set of gates (e.g. AND/OR/NOT or simply NAND) that are "universal" in the sense that they and their compositions are sufficient to represent any function.  While these sets do not affect membership in complexity classes such as NC or AC, they affect scaling prefactors and thus could have practical implications for network implementations.  Similarly, in the present case, the difference between neurons that can represent functions of greater or lesser complexity can and will have an impact on the number of neurons necessary for a circuit to implement some function.  These results thus have a direct consequence on the size of networks in both artificial and biological systems necessary to implement given functions.
>
> We understand the concern that real learning happens in circuits, not only in isolated neurons. However, we believe it is still important to understand what individual neurons can and cannot compute based on their morphology (e.g. Beniaguev et al Cell 2021 mentioned by reviewer 2).  Many cortical neurons differ significantly in their dendritic structures, and this variation likely constrains what role they can play within a network.
>
> Our goal is not to suggest that neurons act merely as standalone classifiers. Instead, we are asking what kinds of computations a neuron is structurally capable of performing. We show, for example, that some architectures support broader function classes but may generalize poorly, while others show the reverse pattern. This kind of tradeoff could be important for understanding how neurons specialize or adapt during learning. We have added a short discussion of this point in the revised manuscript to make our scope clearer.
>
> Thanks again for your thoughtful feedback. Your comments helped us sharpen the framing of the paper, and we have revised the manuscript to reflect those clarifications.

---

> > ### Comment · Reviewer_BeVr · 2025-08-03
> >
> > I appreciate the detailed responses provided by the authors. However, I remain unconvinced that the very low dimensionality used in their analysis is well justified, and therefore I will maintain my current evaluation.

---

### Official Review · Reviewer_j1t9 · 2025-07-01

**Clarity:** 3
**Significance:** 4
**Originality:** 4
**Rating:** 5
**Confidence:** 4

**Summary:**

This paper computationally examines how a simplified model of neuron morphology can impact the types of functions it can learn. The authors show that "basal" like neurons learn a range of functions, however are not able to generalize well, while "apical" like neurons can quickly generalize, but learn less reliably. These results provide insight into why the brain has both apical and basal dendrites.

**Questions:**

I do not have any questions for the authors.

**Ethical Concerns:**

["NO or VERY MINOR ethics concerns only"]

**Final Justification:**

I really enjoyed this paper and I think it is of high quality and will be of interest to the NeurIPS community. For those reasons, I think it should be accepted.

**Limitations:**

The authors nicely addressed their limitations in the Discussion section.

**Quality:**

4

**Strengths And Weaknesses:**

**STRENGTHS**

1. This paper is well motivated with a clear Introduction and Abstract.

2. Figure 1 is very clear and really helps the reader understand the methods of the paper.

3. The paper was well scoped - it took on just the right amount enabling a clear result that will be of interest to the NeurIPS community.

4. The re-training results are quite interesting and suggest and interesting trade-off.

5. I love the summary "morphology sculpts computation, shaping not only what neurons do, but how they learn and adapt". I think that is a really great conclusion of the authors work (and could possibly even be added to the Abstract). Table 1 is also a helpful summary.

6. The implications for deep learning are interesting and I enjoyed that they were commented on.


**WEAKNESSES**

1. The first 3 sections of the paper were (largely) very clear and well written. The main weakness of this paper is that the next sections feel a little more rushed and less clear, particularly Section 5. I think there are a few straightforward things that could be done to address this:
    a. While I appreciate that Fig. 3C illustrates the authors' point about the difference between apical and basal dendrites based on how well they learn functions of different sensitivities, I feel like there might be a better way to show this. Possibly is matrix heat map, where each square is the mean sensitivity for each of the different pairwise probability values (I know that these values varied, but the plot currently already has a block like structure, so taking the average across all the dots in the block and plotting in a heat map would make it easier to read).
    b. Fig. 4 should be made bigger. It's hard to read as is and I think makes it much harder to grasp the authors' point about the difference between the neuron types and their re-training abilities. This could be achieved by moving the legend of Fig. 4C underneath the plot (instead of to the side) so that the figure can be scaled larger.
    c. I think the Discussion section (which is well written) could be strengthened by more citations/connection to existing literature (especially the neuroscience literature).

2. This work cites relatively few other papers. While I understand the preference of not citing things that are not essential to the authors' work, I think the lack of citations makes it harder for the reader to understand exactly how this work situates itself within the existing literature. Especially commenting on more neuroscience literature would be helpful - there is work studying basal and apical dendrites and their computations. How does this work compare? How might a neuroscience experiment test the authors' conclusions? What implications does this work have for studies showing that basal and apical dendrites in hippocampus are not stable? (These are just a few thoughts that came to mind - no need to address any or all of them, but thinking some more about some connection I think would be good). I was also surprised Beniaguev et al. (Cell 2021) was not cited in the Introduction when discussing the computational roles of dendrites. It's my understanding that that is a landmark paper that has shaped how people in the community think about the computational effect of dendrites.

**MINOR POINTS**

1. I think this work would benefit from a careful re-read to catch typos (there are several instances where there are no spaces between words) and inconsistencies in how equations and figures are referenced. For instance, Figure 1 a, Figure 1B, Figure 1 B, Figure2B, Supplemental figure 6, and 1(a) are all used to refer to specific figures. Similarly, Equation 7 and 5 are used to refer to specific equations. Making it more consistent would help the reader (although, I recognize this is a more minor point).

---

> ### Author Rebuttal · Authors · 2025-07-31
>
> Thank you very much for the thoughtful and generous review. We are glad that the core message  of the paper, "morphology sculpts computation, shaping not only what neurons do, but how they learn and adapt" , resonated with you, and we have added this line to the abstract as you suggested. Your comments on clarity, scope, and implications for both neuroscience and ML were encouraging, and we've aimed to improve the manuscript based on your specific suggestions.
>
> ## Clarity of Figures (Fig. 3C and 4)
>
> We appreciate your detailed suggestions here. We will revise Figure 3C to show a matrix-style heatmap that summarizes the average sensitivity for each input probability pair. This more compact format highlights the structure that was already implicit in the scatter-style plot and makes the architectural differences easier to interpret at a glance.
>
> For Figure 4, we will increase the size of all panels and adjust the layout to improve readability. In particular, we will move the legend below the plot and expand the axes labels. These changes should help clarify how different architectures behave during re-training.
>
> ## Citations and Neuroscience Context
> We completely agree with your point that the paper could better acknowledge relevant neuroscience work,  especially Beniaguev et al. (Cell, 2021). This was actually one of the key papers that inspired us to think about single neurons not just as shallow units with one hidden layer as previously thought,  but potentially having a deeper architecture. While Beniaguev et al also focus on the role of receptors in increasing the effective depth of the single-neuron ANN, we focused on a simplified model that brought into focus the intrinsic depth vs breadth tradeoff in dendritic computation that arises due to the relative arrangement of a limited number of non-linearly integrating dendritic compartments - thus enabling the exploration of single-neuron dendritic integration in terms of computational complexity.  We regret not citing it earlier and have now included a discussion of its relevance in the Introduction and Discussion sections.
>
> In the revision, we also plan to highlight several recent experimental and theoretical works that inspired, and support, our framing. For example, Otor et al. (2022) demonstrate that layer 5 tuft dendrites exhibit morphologically defined compartmentalized computations during motor tasks. Their findings show that dendritic structure and NMDA-based nonlinearity shape how distinct motor variables are represented within the same dendritic tree - supporting the idea that morphology constrains local computation. Similarly, Gidon et al. (2020) discovered graded calcium-mediated dendritic spikes in human L2/3 pyramidal neurons, enabling single neurons to solve linearly non-separable tasks typically requiring multilayer networks. These studies reinforce the growing consensus that dendrites contribute substantially to single-neuron computational power.
>
> We also appreciated the reviewer’s comment to consider hippocampal work. Recent studies by Li et al. (2023) and O’Hare et al. (2025) show that distal dendrites in hippocampal CA3 and CA1 neurons contribute to place field formation and memory flexibility, and are dynamically recruited during learning. These findings support the idea that dendritic subregions encode functionally distinct computations, possibly shaped by morphology and context. Building on our proposed framework, it could be useful to compare different cell types across brain regions in terms of their computational complexity, to understand whether region-specific cell types emerged to accomodate certain computational demands.
>
> We also plan to cite recent reviews, such as Tye et al. (2024) which discusses how nonlinear mixed selectivity increases representational dimensionality. Their proposed mechanism - selective gating of variable mixtures echoes our architectural framing, where local nonlinear units gate combinations of inputs, effectively shaping inductive biases and generalization capacity. Finally, we acknowledge a broader body of modeling work (e.g., Kim et al., 2023, work by Poirazi group, Mel Group, Naud Group) that explores advanced computations enabled by dendritic trees through models of dendritic integration.
>
> Given recent progress in in-vivo dendritic imaging, we foresee ample opportunity to test conclusions from this kind of analysis. For example, Aggarwal et al. (2023) introduced a method to track changes in synaptic activity in different dendritic compartments alongside somatic output. Techniques like this could help address questions such as:
>
> - Do apical and basal dendrites show different learning dynamics during behavior?
> - Do they support different levels of generalization across tasks?
> - Are there differences in how synapses are rewired across dendritic domains during learning?
>
> We plan to add a short section to the Discussion outlining these directions, and hope they’ll help bridge the gap between the theory we present and emerging experimental work.
>
> ## Minor Fixes and Formatting
> We have gone through the paper to fix:
> - Inconsistent figure and equation references (e.g., “Figure 1a” vs. “Fig. 1A”).
> - Typos, formatting issues, and spacing.
> - Some unclear or crowded figure captions.
>
> Thanks again for your thoughtful and constructive review. Your suggestions helped us strengthen both the framing and clarity of the paper, and we are hopeful that the revised version does a better job of connecting with both ML and neuroscience readers.

---

> > ### Comment · Reviewer_j1t9 · 2025-08-01
> >
> > I thank the authors for their detailed rebuttal. All my questions have been addressed. I think the discussion on the connection to the neuroscience literature (especially the highlighted questions motivated by the authors' work that experimentalists should examine) is great!

---

### Official Review · Reviewer_C15H · 2025-07-02

**Clarity:** 2
**Significance:** 2
**Originality:** 3
**Rating:** 4
**Confidence:** 3

**Summary:**

This paper considers the computational ability of single neurons, which are modeled as simple feedforward relu networks. Both deep/narrow and wide/broad network topologies are considered for the neurons dendrites, each of which have different pros and cons, highlighting that the morphology of neurons leads to different functionality at the level of single neurons.

**Questions:**

The biggest question I had was about the training protocol used, which is not clearly outlined either in the main text or supplemental. I would recommend replacing comments about training times with clear outlines on how these networks are trained. This is especially important here, as it is not obvious from the text if the network fails to learn a particular boolean function because the network architecture does not support it, or if it is because of a failure due to the learning protocol (due to, say, a nonconvex learning landscape). I believe this is why the authors also include results for best case realizability, it would be good to make this clear in the text. Even with multiple learning trials for a single function, there are situations where the learning protocol may never be able to realize a function that is still possible to implement by the network. I think it would be good to make this limitation a bit more clear.

This paper also effectively targets questions about functional realizability in terms of network breadth and depth. There is a bit of literature on this topic, see for example M. Telgarsky (2016) or D. Rolnick & M. Tegmark (2018). Though I'm not sure if these results are immediately applicable, they might be a good direction to look into.

Some smaller nitpicks:
* What is being shown in figure 3B could be made more clear in the description
* The text in figure 4 is very hard to make out
* Although a bit beyond the scope of this paper, even some cursory remarks on the effect of local, neuronal level, functional realizability on the behaviour of larger networks I think would make this paper more interesting to the larger NeurIps community.

Not so much a question for this paper, but there a lot of interesting directions to take this type of study.
Along with the many directions listed in the discussion, it would be interesting to explore how this extends to neurons with multiplicative presynaptic interactions, such as those that can occur during presynaptic gating, as such interactions are fundamentally different then the additive ones used in standard ANNs but quite relevant for biologically plausible organizations

**Ethical Concerns:**

["NO or VERY MINOR ethics concerns only"]

**Final Justification:**

I have increased my score to borderline accept, as I believe the authors addressed my main concerns in their rebuttal about the training protocol. However, I believe this work remains limited in scope by addressing only individual neuronal ability, and not effects at the level of networks of basal or apical like neurons.

**Limitations:**

Discussed in strengths and weaknesses and questions

**Quality:**

3

**Strengths And Weaknesses:**

Strengths:
* Individual neuron complexity, and how it effects computation in neural networks, is theoretically understudied and an interesting research direction
* The problem motivation and approach are clearly written and easy to understand.
* The figures are mostly clear and easy to read
* Limitations are clearly laid out, and leave a lot of open directions

Weaknesses:
* The biggest weakness is in the clarity and possible robustness of the training (see questions below)
* They discuss effects of neuron morphology on the level of single neuron computation, but it is not clear in what way these morphologies impact computation at the level of networks of many neurons. Making stronger connections between single cell computational ability and the capabilities of networks composed of these neurons would strengthen this work.

---

> ### Author Rebuttal · Authors · 2025-07-31
>
> We thank the reviewer for the constructive feedback. We are glad that the motivation and clarity of the manuscript came through, and appreciate the reviewer’s interest in the broader significance of our findings. We address the weaknesses and questions listed by the reviewer below:
>
> ## Training protocol
> Thank you for the questions about our training protocol.  We apologize for the lack of detail.  We will follow your suggestion and add clear statements about training methods, which we detail below.  It is important to note that we used two distinct approaches, gradient descent, as well as a non-gradient based method (DIRECT) in order to test for limitations of the learning algorithms.
>
> For all training instances, which entailed learning the function mapping $f : \\{0,1\\}^n \rightarrow \\{0,1\\}$ in input dimension $n$, the input consisted of the full enumeration of $2^n$ input rows, with the target being the Y-column of the boolean function mapping.
> While training using the gradient-descent based approach, we used Binary Cross-Entropy as the loss function between the predicted and the target outputs during the training. We used an Adam optimizer with a fixed learning rate, with learning rates and epochs tuned according to the input dimension to ensure convergence (ranging from 6e4 epochs with lr = 5e-3 for $n_{dim} =  3$ to 6e6 epochs, lr =  1e-4 for $n_{dim} =8$). We also ran instances of training with an Adam optimizer with a learning rate scheduler, but that did not impact realizability trends shown in the paper.
>
> For the DIviding RECTangles (DIRECT) optimization, we used Scipy's implementation which was based on Gablonsky et al 2001. Reasonable bounds on the learnable parameters were first set as input parameters for the optimization algorithm. The parameter space was constrained with weights bounded between -20 and 20, branch gains ($h$) between 0 and 10, thresholds ($\theta$) between -1 and 10, and output bias ($\Delta$) between -1 and 10. Both basal and apical cell architectures were trained using mean squared error loss between continuous sigmoid outputs and binary target labels. The DIRECT algorithm performed global optimization without gradients, exploring the 7*d +7 + 7 + 1 - dimensional parameter space (d input dimensions × 7 branches + 7 gains + 7 thresholds + 1 bias) to minimize classification error with a maximum of 2,000,000 function evaluations and zero volume tolerance for convergence. Training performance was evaluated using Hamming distance between binarized predictions (threshold 0.5) and ground truth binary labels.
>
> For a broad overview on the evaluation of realizability of a boolean function by a given architecture, we point the reviewer to Supplemental Figure 1.
>
> We agree that distinguishing between realizability and trainability is crucial. To clarify, our experiments include both “trainable realizability” (via supervised gradient descent or DIRECT optimization (Gablonsky J et al, 2001) and “best-case realizability”. In the revised version, we will explicitly describe our training setup in the main text, including the number of trials, optimizer used, convergence criteria, and the role of DIRECT as an alternative to backpropagation to confirm that the phase transition in realizability is independent of the training algorithm.
>
> As the reviewer notes, some realizable functions may still be difficult to learn due to non-convexity or poor conditioning of the loss landscape. We agree and have made this distinction clearer in both the Results and Limitations sections. We believe that this also reflects a biologically-relevant constraint: real neurons must compute within the bounds of both their physical structure that impose constraints on parameter regimes and resolution. Our empirical results are thus best interpreted as a combination of architectural capacity and ease of access through plausible optimization, a point we now emphasize more clearly.
>
>
> ## Network-level implications
>
> Thank you for the suggestion to discuss the network-level implications of this work.  While we focus on single neuron computational complexity, these results have immediate impact on the complexity of entire circuits.  A similar question arises in the domain of circuit complexity when considering Boolean circuits.  In that case, one specifies a set of gates (e.g. AND/OR/NOT or simply NAND) that are "universal" in the sense that they and their compositions are sufficient to represent any function.  While these sets do not affect membership in complexity classes such as NC or AC, they affect scaling prefactors and thus could have practical implications for network implementations.  Similarly, in the present case, the difference between neurons that can represent functions of greater or lesser complexity can and will have an impact on the number of neurons necessary for a circuit to implement some function.  These results thus have a direct consequence on the size of networks in both artificial and biological systems necessary to implement given functions.
>
> ## Minor points
> - We have clarified the description of Figure 3B and improved the readability of Figure 4.
> - We will also cite and briefly comment on work by Telgarsky (2016) and Rolnick and Tegmark (2018), and note how our setting differs (e.g., binary function realizability, fixed input dimension).
> - The reviewer’s point about multiplicative interactions and gating is well-taken. As proposed in the review Payeur et al (2019), more complex models could be used to model such interactions.  While outside our current scope, we agree this is a promising direction, and have included it in our extended discussion.

---

> > ### Comment · Reviewer_C15H · 2025-08-08
> >
> > Thank you for clarifying the training protocol, which had addressed my main questions involving the training. Although I believe that the lack of the effect of neuron morphology on network level behaviour remains a blind spot for this work, I have updated my score in light of this detailed response.

---

### Note · Authors · 2025-08-12

We thank all reviewers for their careful reading of our paper and their thoughtful engagement during the discussion phase.

We are grateful for nE68's observation that "The work opens up new directions for incorporating biologically grounded neuron models into machine learning, particularly for tasks involving continual adaptation and resource efficiency."

j1t9 highlighted the paper’s clear motivation, well-scoped experiments, implications for deep learning, and final takeaways. We’re grateful they highlighted the summary “morphology sculpts computation, shaping not only what neurons do, but how they learn and adapt,” which we'll add to the abstract. They also noted that the Discussion is well written; per suggestion, we will add more connections to the neuroscience literature.

We’re pleased that C15H’s main concern about the training protocol was addressed.  Regarding the broader relevance to network-level behavior, we will clarify how our retraining experiments, using branch-specific weight freezing, mirror mechanisms like dendritic disinhibition, which has been shown to gate plasticity (Bustos et al., 2022) and support context-dependent learning (Onasch et al., 2025). Our results suggest that dendritic architecture plays a direct role in how neurons contribute to task modularity and memory stability at the network level.

We also thank qnNj for prompting a broader discussion on the relevance of our work to modern deep learning architectures. Inspired by their comments, we conducted preliminary experiments applying our architectural motifs to transformer attention heads. These showed that broad (basal-like) architectures tend to achieve better task performance, while hierarchical (apical-like) architectures learn faster and show stronger transfer across tasks. Similarly, BeVr noted that "The distinction between apical and basal architectures is an important contribution."  These results reinforce the broader value of dendritic-inspired structure in artificial systems, and we plan to include them as a short section or appendix to highlight the NeuroAI relevance of our work.

Planned revisions include:
 - Clearer exposition of training and function sampling methods
- More consistent and accessible figure formatting
- New citations to hippocampal dendritic studies and broader neuroscience literature
- Transformer-based analysis showing NeuroAI applications of our approach

---

### Decision · Program_Chairs · 2025-09-17

**Decision:**

Accept (poster)

**Comment:**

This work studies the computational power of single neurons as a function of neuronal morphology, where the neuron is seen as a simple network of linear threshold units. The paper finds a learnability threshold for random Boolean functions in terms of the number of inputs of the function, which depends on the size of the dendritic tree. Moreover, the paper identifies two classes of morphologies ('apical' and 'basal') that can learn disjoint sets of functions and that exhibit different learning and generalization behavior.

Reviewers found the topic and the results interesting and within scope for the conference. Initial concerns focused on a general lack of clarity in the manuscript (regarding especially the definition of the classes of functions defined in the work and the details of the analysis) and the rather limited dimensionality (N≤8) of the functions tested. During the rebuttal and discussion period, the authors clarified several confusion points, improved their discussion of prior work, and argued for the rationale behind their use of low-dimensional inputs. Moreover, one of the reviewers requested some exploration of ways in which this study could inform the design of real-world ANNs, which led to some preliminary experiments with transformers hinting at a conceptual agreement with the core results.

Overall, the paper explores a somewhat understudied topic with an original approach; the results are solid, and interesting connections are drawn to broader problems more directly applicable to practical machine learning. Despite some scores remaining borderline, this work in my opinion meets the criteria for acceptance.